# Shifting the Immune-Suppressive to Predominant Immune-Stimulatory Radiation Effects by SBRT-PArtial Tumor Irradiation Targeting HYpoxic Segment (SBRT-PATHY)

**DOI:** 10.3390/cancers13010050

**Published:** 2020-12-26

**Authors:** Slavisa Tubin, Seema Gupta, Michael Grusch, Helmuth H. Popper, Luka Brcic, Martin L. Ashdown, Samir N. Khleif, Barbara Peter-Vörösmarty, Martin Hyden, Simone Negrini, Piero Fossati, Eugen Hug

**Affiliations:** 1MedAustron Ion Therapy Center, Marie Curie-Straße 5, 2700 Wiener Neustadt, Austria; piero.fossati@medaustron.at (P.F.); Eugen.hug@medaustron.at (E.H.); 2Lombardi Comprehensive Cancer Center, Georgetown University Medical Center, Washington, DC 20057, USA; sgjija@gmail.com (S.G.); snk48@georgetown.edu (S.N.K.); 3Institute of Cancer Research, Department of Medicine I, Medical University of Vienna, 1090 Vienna, Austria; michael.grusch@meduniwien.ac.at (M.G.); barbara.peter-voeroesmarty@meduniwien.ac.at (B.P.-V.); 4Diagnostic and Research Institute of Pathology, Medical University of Graz, 8010 Graz, Austria; helmut.popper@medunigraz.at (H.H.P.); luka.brcic@medunigraz.at (L.B.); 5Faculty of Medicine, Dentistry and Health Sciences, University of Melbourne, Melbourne 3010, Australia; mlashdown@optusnet.com.au; 6Institute for Pathology, Kabeg Klinikum Klagenfurt, 9020 Klagenfurt am Wörthersee, Austria; martin.hyden@kabeg.at; 7Internal Medicine, Clinical Immunology and Translational Medicine Unit, IRCCS Ospedale Policlinico San Martino, 16132 Genoa, Italy; negrini@unige.it

**Keywords:** immune-microenvironment, partial irradiation, abscopal effect, bystander effect, SBRT, timing

## Abstract

**Simple Summary:**

This review presents and summarizes the key components and outcomes of a novel, unconventional radiation approach aimed to exploit immune-stimulatory radiation effects which, being added to direct radiation tumor cell killing, may improve the therapeutic ratio of radiotherapy. This technique, as a product of translational oncology research, was intentionally developed for the induction of immune-mediated bystander and abscopal effects in the treatment of unresectable bulky tumors which have much fewer therapeutic options and show poor prognoses after conventional treatments. This review offers insights into a unique unconventional radiotherapy technique which, due to its higher immunogenic potential, may improve the prognosis of patients affected by highly complex malignancies, providing additional opportunities for future research in terms of combining novel immuno-modulating agents with more modern radiotherapy approaches.

**Abstract:**

Radiation-induced immune-mediated abscopal effects (AE) of conventional radiotherapy are very rare. Whole-tumor irradiation leads to lymphopenia due to killing of immune cells in the tumor microenvironment, resulting in immunosuppression and weak abscopal potential. This limitation may be overcome by partial tumor irradiation sparing the peritumoral immune-environment, and consequent shifting of immune-suppressive to immune-stimulatory effect. This would improve the radiation-directed tumor cell killing, adding to it a component of immune-mediated killing. Our preclinical findings showed that the high-single-dose irradiation of hypoxic tumor cells generates a stronger bystander effect (BE) and AE than the normoxic cells, suggesting their higher “immunogenic potential”. This led to the development of a novel Stereotactic Body RadioTherapy (SBRT)-based PArtial Tumor irradiation targeting HYpoxic segment (SBRT-PATHY) for induction of the immune-mediated BE and AE. Encouraging SBRT-PATHY-clinical outcomes, together with immunohistochemical and gene-expression analyses of surgically removed abscopal-tumor sites, suggested that delivery of the high-dose radiation to the partial (hypoxic) tumor volume, with optimal timing based on the homeostatic fluctuation of the immune response and sparing the peritumoral immune-environment, would significantly enhance the immune-mediated anti-tumor effects. This review discusses the current evidence on the safety and efficacy of SBRT-PATHY in the treatment of unresectable hypoxic bulky tumors and its bystander and abscopal immunomodulatory potential.

## 1. Introduction

With the remarkable technological developments in the treatment planning and delivery, the modern high-precision, image-guided radiotherapy becomes one of the leading treatment options in cancer management. Stereotactic body radiotherapy (SBRT) delivers an ablative radiation dose to a tumor with millimetric accuracy, exposing some portion of the surrounding healthy tissues to a mid-low radiation dose. SBRT represents the treatment of choice for limited-volume primary and metastatic lesions, especially for the patients who refuse surgery, or for whom surgery is not indicated. Based on the tumor volume and radiation dose used, for most of the treated lesions, local control rates of 80% or higher can be reached with an improvement of the overall survival, even for oligometastatic patients [1,2]. However, these clinical outcomes cannot be reproduced by the patients with high-volume tumors even with most advanced radiotherapy techniques, which is especially true for patients with unresectable bulky tumors. In most of these cases, the delivery of an ablative radiation dose to the whole tumor by means of conventional radiotherapy is limited by surrounding tissue tolerance that makes a curative treatment demanding. Bulky tumors are very challenging to treat not only because of the high volume and intimal relationship with usually infiltrated nearby organs, but also because of the presence of tumor hypoxia determining an adverse prognosis [3]. Additionally, radiotherapy of large-volumes, which is the case for bulky tumors, might potentially bring another unfavorable aspect of radiation-induced lymphopenia.

Radiotherapy has the immunomodulatory potential thereby affecting the dynamics of the immune response. The present evidence on the interaction between the radiation and immune system are controversial, showing that radiation can exert both immune-stimulatory [4,5,6] and immune-suppressive effects [7,8]. The clinical dominance of such radiation-mediated immune effects is determined by the irradiated-volume, dose-fractionation and radiotherapy technique. Clinical data show that the conventional radiotherapy predominantly generates immune-suppressive effects [9]. Furthermore, it has been shown that irradiation of larger volumes in multiple daily fractions correlates with radiation-induced lymphopenia [10,11], leading to a global immunosuppression and consequent poor oncologic outcome [12,13]. These radiation-mediated immune-suppressive effects may negatively affect the therapeutic efficacy in many tumor types [14,15]. This also results in the inhibition of immune-stimulatory effects, consequently blocking the induction of immune-mediated abscopal effects (AE).

Since the 1950s, when it was described for the first time [16], radiation-induced AE has attracted lot of interest in the radiation oncology community due to its therapeutic potential, especially in the last decade-era of immunotherapy. AE (systemic) together with radiation-induced bystander effect (BE, local), represent phenomena of non-targeted (outside the irradiated treatment field) anti-tumor effects of radiation [17]. Practically, this means induction of regression of distant-metastatic (AE) and loco-regional (BE) tumor tissues that were not directly targeted with local radiation. Although AE was a very rare clinical phenomenon in the last six decades, an increasing number of reports have been recorded after the use of SBRT in combination with immunotherapy [18]. Hypothetically, the mechanisms behind AE and BE are “immune-mediated” or “cytokines-based” [19], being cellular and soluble mediators probably both involved. AEs have been sporadically documented, especially following hypo-fractionated high-dose radiotherapy while BEs have not been reported in the clinic following the conventional whole-tumor irradiation. First experimental evidence on BEs appeared in 1990s [20]. Since then, exclusively unconventional, “spatially-fractionated” approaches like SBRT-based PArtial Tumor irradiation targeting HYpoxic tumor segment (SBRT-PATHY), GRID or LATTICE brought BEs to the clinic [21,22,23,24]. Particularly, these techniques expose only some parts of the tumor (and not total tumor) to the high-ablative radiation dose sparing the peritumoral tissue which is compatible with triggering the mechanisms responsible for BE-induction.

The objective of this review is to explore SBRT-PATHY model of partial tumor irradiation purposefully developed to spare the peritumoral immune-microenvironment (PIM) as an “organ at risk” (OAR) from radiation. This would shift the predominant radiation-induced immune-suppressive effect of the conventional whole-tumor irradiation into immune-stimulatory effect, leading to therapeutic advantage facilitated by BE and AE-mediated tumor cell killing. Furthermore, the available evidence on this long-standing translational oncology research, including the preclinical basis, clinical outcomes of SBRT-PATHY in the treatment of unresectable, hypoxic, bulky disease, as well as the immunohistochemistry (IHC) and gene-expression findings, supporting an immune-stimulatory role of PIM, will be discussed. The concept follows the hypothesis that for induction of prevalent immune-stimulatory radiation effect partial tumor irradiation sparing PIM is required in order to improve the therapeutic ratio by adding to the exclusive radiation-directed, also an immune-directed tumor cell killing. PIM, containing the loco-regional antigen-presenting cells (APC) and circulating lymphocytes, as well as a source of the pro-inflammatory cytokines, should be preserved, intact and functional. This review will therefore summarize the translation of our preclinical findings to the clinic that has resulted in the development of an effective, unique, and novel approach for the induction of the BE and AE.

## 2. Translational Oncology Research

### 2.1. Partial Tumor Irradiation

Partial tumor irradiation, as an unconventional alternative to the traditional whole tumor-volume radiation, might offer potential benefits for those tumor entities that can’t be optimally treated with standard approaches. As previously mentioned, unresectable bulky tumors are characterized by the very large volume, significant amount of radio-chemo-resistant hypoxic tumor cells and intimal relationship with the nearby critical organs which reduce the radiation dose prescription to the sub-tumoricidal, usually palliative level. All these together contribute to the conventional treatment failure. Therefore, partial tumor irradiation might be an option to deliver a higher radiation dose to the tumor in a safe way without increasing the risk for detrimental OAR-damage, including the peritumoral immune system cells usually exposed to full radiation dose by the conventional approach. The rationale behind this approach is that an immunogenic high-dose radiation (that otherwise couldn´t be delivered to the whole tumor volume), even if delivered only to a part of the tumor, might be enough for an effective immune modulation in order to release (hidden) tumor neoantigens and adjuvant activating signals, leading finally to activation of the radiation-spared peritumoral immune environment. This would mean to add an immune-mediated whole-tumor killing component to the direct radiation-mediated partial-tumor killing component.

### 2.2. Preclinical Findings

It has been shown that partial tumor irradiation is able to induce AE and BE and anti-tumor immune responses. The significant bystander cytotoxic killing has been observed in “shielded” unirradiated tumor cells located nearby the high-dose radiated regions with GRID therapy [25]. GRID technique delivers high radiation dose through a perforated screen with blocked areas called a “GRID” [26]. This effect, specific for GRID-partial tumor irradiation, is not observed with conventional radiotherapy which encompasses the entire tumor volume. It has been shown that this tumoricidal BE is mediated by TNF-α and TRAIL, secreted by the directly irradiated tumor cells [27,28,29,30]. The studies in murine tumor models demonstrated that partial tumor irradiation is responsible for an increased immune-mediated tumor cell death in unirradiated distant tumor compared to the whole tumor irradiation [31]. In a syngeneic mice tumor model, partial tumor volume irradiation using LATTICE therapy (20 Gy) induced increased pro-inflammatory cytokines level with consequent T-cell infiltration, leading to delayed tumor growth of unirradiated distant tumor [32]. Finally, investigators from Memorial Sloan Kettering Cancer Center reported anti-tumor responses after partial irradiation in 67 NR murine orthotopic breast tumors [33]. Here, single dose of 10 Gy delivered to half of the tumor led to reproducibly inducible antitumor immune responses that eliminated the entire tumor in immunocompetent mice, but not in nude mice. Additionally, a significant AE was observed in the unirradiated tumors.

Although, the above-mentioned pre-clinical studies utilized partial tumor irradiation, the selection of the tumor region for irradiation remained random. It is very well known that most of the solid tumors have a hypoxic tumor region that is radioresistant [34]. Bulky tumors, in general, have larger hypoxic regions, contributing to the ineffectiveness of radiation therapy for such tumors [35]. The role of this hypoxic region in generating an AE/BE has not been explored. Therefore, we planned an in vitro study starting in 2010 to evaluate this aspect. Following the hypothesis behind the partial tumor irradiation, the objective of this preclinical study was the exploration and analysis of the tumor subvolumes to define the “most-abscopal one”. To the best of our knowledge, this was the first experimental study investigating the impact of irradiation of hypoxic cancer cells on their growth in terms of both induction of AE (generation in the presence of hypoxia) and its effect on the hypoxic tumor cells [36]. For that purpose, two non-small cell lung cancer (NSCLC) cell lines, A549 and H460 were exposed either to hypoxia or normoxia. After the selection of the hypoxia-resistant clones (HR: A549/HR, H460/HR), all cell groups (A549, A549/HR, H460 and H460/HR) were either left untreated or irradiated with conventional radiation dose of 2 Gy- or an ablative dose of 10 Gy-single fraction. Two different radiation dose-levels were used in order to evaluate the dose-dependence of AE-intensity. After 24 h, unirradiated hypoxic (H-CM) or normoxic (N-CM) conditioned media (CM), and irradiated hypoxic (H-RCM) or normoxic (N-RCM) CM were obtained and used for subsequent experiments. Unirradiated parental cells or HR clones were then exposed to H-CM, N-CM, H-RCM or N-RCM and cell growth and proliferation were continuously monitored and quantified by real-time cell electronic sensing system. In addition to these non-targeted “abscopal” (media-transferred) radiation effects, also the effects of combined direct-indirect radiation were explored by exposing in parallel the same set of cells to direct irradiation of 2 Gy 24 h after their incubation with CMs to evaluate the radio-sensitizing potential of the “abscopal cytokines”. Levels of hypoxia and HIF1α regulated angiogenesis related growth factors, basic fibroblast growth factor (bFGF), placental growth factor (PlGF), soluble fms-like tyrosine kinase (sFlt-1) and vascular endothelial growth factor (VEGF) were assessed by electrochemiluminescence detection in each sample of CM or RCM.

The results from this study revealed some very interesting and novel findings. We found that the final non-targeted effect of irradiation is not a constant, but is determined by the balance between several factors, including the tumor cell type and differentiation grade (both inducers and recipients of AE), inductive radiation dose, microenvironmental oxygen status (normoxic vs. hypoxic), and effect of released cytokines. Some treatment conditions resulted in very strong anti(-tumor)-proliferative AEs, for example the high-dose 10 Gy-single fraction irradiation of the hypoxic, radio-resistant tumor cells as inductor and receiver. On the other hand, the conventional-dose 2 Gy-single fraction irradiation of the normoxic, radio-resistant tumor cells as inductor and receiver resulted in completaly oposite-stimulative, pro(-tumor)-proliferative AE. Thus, the non-targeted effects of radiation might be significantly different in terms of intensity and/or type depending on which region of the tumor is targeted: hypoxic versus normoxic versus both (whole tumor). Strong anti-proliferative AE were observed under hypoxia (H-CM, H-RCM) compared to normoxia (N-CM, N-RCM). Further, “radiation-hypoxia-induced AE” also improved the radio-sensitivity of radio-resistant cancer cells, including the hypoxic cells. These findings demonstrated for the first time that the selective irradiation of hypoxic tumor cells as inductor of the non-targeted effects, resulted in significant AEs. Finally, the comparative analysis of the growth factor levels in RCM and CM with cell growth showed a correlation between anti-proliferative sFlt-1 and almost all RCM and HCM types for both the cell lines, indicating its important role in mediating AE.

### 2.3. Translation of the Preclinical Findings to the Clinic

Since the preclinical findings identified the hypoxic tumor cells as potentially more “AE-inducing” than the normoxic cells, hypoxic tumor segment was selected as a potential target-inductor of BE and AE for the clinical purpose. The novel therapeutic concept implied the identification of the hypoxic tumor subvolume and its subsequent irradiation initially with a single fraction of 10 Gy prescribed to the 70% isodose line in order to increase radiation dose within that hypoxic segment up to 14.5 Gy. With this approach, the hypoxic tumor segment was exposed to a dishomogenous radiation dose ranging from 10 Gy (in its periphery) up to 14.5 Gy (in its center) which, considering the evidence on most immunogenic radiation dose [36,37,38], would correspond to an optimal “immunogenic dose” increasing the probability of abscopal response. First, seven patients were treated starting in 2016 [21]. The treatment was performed with SBRT-volumetric modulated arc therapy (VMAT) technique in order to maximize the radiotherapy precision and to minimize radiation dose outside the tumor. The selected patients were affected by the symptomatic, unresectable bulky tumors that were progressive under previously recommended state of the art treatments per stage of the disease, including either systemic therapy and/or conventional radiotherapy. After discussing the unconventional nature of the partial tumor irradiation and associated potential risks, all patients signed the informed consent and accepted this treatment with the hope to control the disease-related symptoms. All clinical studies were conducted after approval by Institutional Review Board and all procedures performed were in accordance with the ethical standards. The studies have been registered by the local ethic committee. Because of the lack of access to hypoxia-specific PET tracer or other more specific hypoxia-imaging techniques at that time, a combination of contrast-enhanced CT and 18F-FDG-PET have been used to define the hypometabolic and hypovascularized tumor segment as the target for SBRT-PATHY. A detailed description of the method can be found elsewhere [21,39].

## 3. Clinical Outcomes

Clinical evidence on the use of this novel approach is still limited to mainly retrospective or non-randomized prospective study data, with small number of patients. Since 2016, 89 patients were treated with SBRT-PATHY concept (Table 1). Sixty-one patients were analyzed retrospectively [21,39,40,41,42], while 28 prospectively [43,44]. All the patients were affected by the unresectable bulky tumors for which an conventional radiotherapy-chemotherapy approach was deemed unsuitable due to tumor volume and site [21,39,41,42] or because of local recurrence within previously irradiated treatment field [40,42]. The male or female patients older than 18 years underwent SBRT-PATHY for solid “bulky” malignancies located in the chest, the abdomen, the pelvis, extremities, brain, or the head and neck region, with limited treatment options. Significant percent of them were treated after being progressed on systemic therapy. “Bulky” disease was considered as a substantial, unresectable tumor mass larger than 6 cm on diagnostic imaging (range: 6–26 cm). The treated tumors encompassed a wide range of malignancies including adenocarcinoma and squamous-cell carcinoma of the lung, malignant melanoma, sarcomas, pancreatic adenocarcinoma, renal cell carcinoma, prostate adenocarcinoma, lymph node-, soft tissue-, adrenal gland- and bone-metastases, chordoma, and head and neck primary and secondary tumors. Most adopted radiation dose was 10 Gy × 3, followed by 10–12 Gy × 1 and 12 Gy × 3, and was prescribed, on an average, to the 30% of the total bulky tumor mass (approximately, 1/3 was targeted for irradiation). The symptoms related to the presence of bulky disease included dyspnea, hemoptysis, pain, cough, dysphagia, bleeding, edema-extremities, and dysphonia. The symptom relief was achieved on an average in 89% of patients after three weeks (range: 2–4). The treatment was well tolerated since only very small proportion of patients presented fatigue-grade 1 and no other side effects. There were no recorded radiation-induced leucopenia.

Considering the available data, with a median follow up of nine months (range: 1–27), the reported median local control was 84% (range: 67–100%). In a substantial number of patients (median 44.5%, range: 28.6–52%), a regional AE was radiographically confirmed [21,39,41,43,44] (Figure 1A–E; an example of SBRT-PATHY of bone metastasis of primary breast cancer). In addition to that, in four patients, an abscopal response was established by the pathological examination.

## 4. Immunohistochemistry and Gene-Expression Findings Following SBRT-PATHY

The main immunohistochemistry and gene-expression findings are summarized in Table 2.

### 4.1. Immunohistochemistry

Four patient-responders to SBRT-PATHY, in terms of radiographically proven significant BE and AE, were submitted to immunohistochemistry and gene-expression analysis. Selected patients had initially unresectable bulky tumors of the lung (one with squamous cell cancer (SCC) and other with adenocarcinoma (AC)) and rectum (both with AC) that were prospectively partially irradiated with neoadjuvant SBRT-PATHY (3 × 10 Gy to 70%; Dmax 43.5 Gy). In addition to primary bulky tumors, all cases presented multiple metastatic regional (mediastinal and pelvic, respectively) lymph nodes. Additionally, the patient with lung SCC presented another primary lung AC in a separate lobe, and one rectal cancer patient had another primary colon AC. Tumor sites other than bulky were not irradiated but followed for the assessment of AE. None of the patients received any systemic therapy. After restaging-CT at 1 month revealed significant BE and AE, patients were submitted to surgery in order to extract responding partially irradiated bulky tumor, radiation-spared PIM and unirradiated abscopal tumor lesions for immunohistochemistry and gene expression analysis.

After standard fixation and preparation of material, slides were stained with hematoxylin-eosin, as well as with antibodies against: CD20, CD3, CD8, CD4, CD56, S100, CD14, CD15, Fox P3, PD-L1 (clone SP263), and apoptosis inducing factor (AIF), using validated protocols.

First case was a patient with unresectable lung SCC, with a separate 2 cm large lesion in another lobe at the same side and lymph node metastases. Restaging CT scan demonstrated significant response of bulky tumor mass (60% reduction), separate lesion in another lobe (50% reduction) and in metastatic lymph nodes (30% reduction). Following surgery, pathohistological analysis demonstrated pronounced necrotic areas in both lesions, partially irradiated SCC, as well as in unirradiated separate lesion in another lobe. In both tumors, there were only around 20% vital tumor cells. The only difference was a dense aggregation of lymphocytes in the PIM-region at the border of necrosis in SCC, which was absent in AC. Lymph nodes showed necrosis, without viable tumor tissue. Immunohistochemistry revealed focal accumulation of CD20+ B-lymphocytes around SCC and some in lymph nodes, while CD3+ T-lymphocytes showed dense infiltration within the SCC and were prevalent in the metastatic lymph nodes (41). The majority of lymphocytes were CD8+ cytotoxic types. On the other hand, CD20+ B-lymphocytes were absent in AC, but some CD3+ T-lymphocytes were present, predominantly CD4+ T-cells. SCC and lymph nodes showed more CD14+ myeloid-derived suppressor cells (MDSCs) in comparison to AC. Samples from all locations presented with small numbers of CD15+ MDSCs. Interestingly, AIF was highly expressed in all three investigated tumor sites. S100 stained macrophages, while CD56+NK cells were not detectable.

Second patient had lung AC. We have received small biopsy samples (prior SBRT-PATHY) as well as resected tumor samples (post-SBRT-PATHY) after the treatment was done. In resection material there were no vital tumor tissue. Large necrotic areas were present, with dense lymphocytic infiltrates in PIM-region at the border to lung parenchyma, foamy macrophages and multinucleated giant cells, with occasional cholesterol clefts and fibrosis. Immunohistochemistry showed that practically all lymphocytes were CD3 positive, with a predominance of CD8+ T-lymphocytes (CD4:CD8 ratio 1:2) (Figure 2). By staining with FoxP3, a higher number of positive cells were detected, even forming small cell clusters. CD20 and PD-L1 were completely negative. The findings in a small biopsy demonstrated also that majority of lymphocytes were CD3+, however with an equal distribution of CD4+ and CD8+ cells. FoxP3 also stained isolated cells, while CD20 and PD-L1 were also here negative. The assessed regional lymph nodes showed no more viable tumor cells following SBRT-PATHY of lung AC.

Third and fourth case were patients with rectal AC, where we also analyzed a small (pre-therapy) biopsy and resected tumor (after the therapy) tissues. After SBRT-PATHY, partially treated bulky tumors demonstrated partial response, with approximately around 25% of vital tumor tissue, with large necrotic areas, and also here lymphocytic reaction with focal fibrosis was obvious. Again CD3+ T lymphocytes formed the majority of cells, with a predominance of CD8+ positive cells (CD4:CD8 ratio 1:2) (Figure 3, Figure 4 and Figure 5). However, here there were also CD20+ cells within vital tumor tissue, while FoxP3 was negative. In a small biopsy (prior SBRT-PATHY) there was similar number of CD4+ and CD8+ cells and few cells were also CD20+ and FoxP3+. PD-L1 was in resected material (post-SBRT-PATHY) and in small biopsy (prior SBRT-PATHY) negative in tumor cells, with few inflammatory cells positive. There were similar findings between these two cases in resection material (post-SBRT-PATHY), with the only difference showing more CD4+ T-lymphocytes than CD8+ T-lymphocytes in one, while a predominance of CD8+ positive cells (CD4:CD8 ratio 1:2) in another. Dense round cell-infiltrate showed very strong cytoplasmatic expression of CD14 in PIM-region. In unirradiated metastatic lymph nodes, there were signs of fibrosis with no viable tumor cells.

### 4.2. Transcript Expression of Cell Death Related Signaling Molecules

Since tumor cell death was observed not only in partially-irradiated tumors but also in non-irradiated sites, the expression of some cell death-regulating signaling molecules was further analyzed by real-time PCR of reverse transcribed mRNAs. For that purpose, total RNA was extracted from sections of the paraffin-embedded tissue blocks with the InnuPrep FFPE total RNA kit from Jena Biosciences. Five hundred ng RNA were reverse transcribed with iScript cDNA synthesis kit (Biorad) and qPCRs were run with iTaq Universal SYBR Green Supermix on a Biorad CFX 96 Real-Time System using primers designed for 60–80 bp amplicons spanning exon boundaries wherever possible. The analyzed genes were the apoptosis-inducing factor (AIF), interferon gamma (IFNG), interleukine 6 (IL-6) and the tumor necrosis factor (TNF) family ligands TNF alpha (TNFA) and TNF-related apoptosis-inducing ligand (TRAIL). Samples were from the same four patients analyzed by immunohistochemistry and were separated into partially irradiated tumor (PIT) and abscopal sites (AS) including non-irradiated tumor sites. Relative gene expression levels were calculated with the dCq method normalized to the housekeeping gene GAPDH.

AIF, IL-6 and TNFA were detectable in all samples. TRAIL was undetectable in one PIT and one AS sample and IFNG was undetectable in two PIT and three AS samples. Of the analyzed genes, IL-6 showed the strongest signals followed by AIF and TRAIL (Figure 6). For AIF, IL-6 and TNFA, abscopal sites had higher expression levels compared to the partially irradiated tumors, whereas this tendency was less clear for IFNG which, however, was undetectable in two of the four PIT samples and TRAIL, which showed a high degree of variation in the PIT samples (Figure 6). Overall, these data suggest an abundance of potentially cell death-inducing signals not only in the partially irradiated tumors but even more so in non-irradiated abscopal sites.

## 5. Why Timing of SBRT-PATHY May Be Important to Break Tumor Tolerance

As far back as 1913 it was hypothesized that “Roentgen Therapy” could affect the immune system and elicit “radio-vaccination” effects [45]. In 1978, Hellstrom et al. hypothesized that an effective tumor response to low dose total body irradiation, could be explained by radiation damage to normal lymphocytes rather than its direct anti-tumor effect, leading to even complete tumor regression [46,47]. One year later, original experiments by Stone et al. had shown the potential role of T cells in tumor elimination by focal radiotherapy [48]. Twice as high a dose of radiotherapy was required to exert an equivalent anti-tumor effect in T cell-deficient animals compared to an immunocompetent mouse. Then, in the late 1980s a series of experiments in mouse tumor models published by RJ North et al. had shown that single, “sub-tumoricidal” doses of either radiotherapy or chemotherapy might cause tumor elimination and prolong survival via immune modulation rather than direct tumor cytotoxicity [49]. The curative effect was reliant on the accurate timing of a single, “pulse” therapy on a specific day post-tumor implantation. Further, if the therapy was applied on days earlier or later than the optimal time, the tumor might even progress. In immune incompetent mice there was no curative effect following the “sub-tumoricidal” single doses. Thus, the timing of therapy and an intact immune system was critically needed to therapeutic success via immune modulation and not direct cytotoxic effect on the tumor.

In order to maximize the probability of therapeutic success in terms of BE/AE-induction, recently, we designed a treatment protocol for serial monitoring of immune-system activity and subsequent synchronization of SBRT-PATHY with its most reactive phase. In addition to partial tumor irradiation targeting of the hypoxic segment and sparing of PIM, SBRT-PATHY was delivered in an estimated “right time”. Importantly, the timing of treatment initiation was determined from a two-week serial monitoring schedule consisting of seven blood-tests measuring Hs-CRP, lymphocytes/monocytes ratio (LMR), and LDH, with the aim to detect the patient’s idiosyncratic cyclical immune fluctuations and periodicity. The putative radiotherapy delivery dates were projected forward into either the 3rd or 4th week following the two weeks of serial monitoring. Specifically, the radiotherapy was given in the pre-trough region on those dates [43]. The hypothesis of our group is that radiation-induced cell killing and subsequent tumor antigen release might result in generation of endogenous cytokines, leading to an effective local and systemic immune-mediated tumor elimination if PIM is radiation-spared and radiation delivered “on right time”. Further, cytokine generation might show the potential to produce endogenous therapeutic levels and thus favorably modulate the underlying local tumor-induced immune suppression in order to stimulate antitumor immunity and systemically break tumor tolerance. Our preliminary data showed cyclical immune response fluctuations of regular frequency. The “right” synchronization of SBRT-PATHY with cyclical antitumor immune activity showed promising clinical outcomes in terms of BE/AE-induction [43]. This topic is the subject of further research for our ongoing prospective trial [44].

## 6. Discussion

A significant insight over the last several years has been the realization that the tumor immune suppression is mediated by the tumor microenvironment, including its immune component [50]. This component is known to consist of regulatory T cells (Tregs), MDSCs and certain cytokines [51]. Further, a confounding issue has arisen in that many of these key cytokines that were originally thought to be pro-inflammatory are now also considered to be immune-suppressive, and are namely bimodal (i.e., IL-2, INFs I & II) [52,53,54]. This suggests that radiotherapy-induced cytokine production might be a “double edged sword”. This would also help explain the dichotomous observations of the immune-stimulatory or immune-suppressive effects of radiotherapy [4].

Another important attribute of cytokines to consider is their normal physiologic short half-life (minutes-hours), low concentrations (pg/mL) and spatiotemporal interactions in maintaining local and systemic immune homeostasis [55,56]. Recent published evidence suggests that the suppressed antitumor immune response is dynamically oscillating over approximately seven days repeatedly in such a homeostatic fashion [57,58,59]. Assuming all homeostatic systems do universally oscillate via feedback, it is not unreasonable to speculate that radiotherapy-induced cytokine release might “skew” the immunosuppressive circuitry of radiation-spared PIM to specifically break local tumor tolerance and cascade systemically to deliver BE and AE. However, this cytokine production would have to occur at a specific time so as to sufficiently extend the normal narrow half-life physiologic restrictions. If radiotherapy damages loco-regional lymphocytes and antigen-presenting cells, and is applied over too long a period or at the wrong time then the bimodal opposing function of the cytokines might promote the immunosuppressive status quo. Thus, the immune oscillation would create narrow recurring therapeutic windows. The previously published data suggest that this temporal therapeutic window could be as narrow as a few hours every several days. Consequently, the random application of hypofractionated radiotherapy could accidentally coincide with the repeating narrow “therapeutic window”. Hence, a limited probability of conjunction of fortuitous events potentially explains the paucity of spectacular abscopal responses after radiation therapy reported since 1953 [16].

Consequently, we hypothesize, radiotherapy fractionation must sufficiently complement the underlying/pre-existing cytokine immune dynamics in an informed but restrained manner to achieve the discrete selective enhanced effects, thus tipping the balance of homeostasis in favor of tumor destructive effects and delaying or even overcoming the normal status quo of suppression/regulation.

Considering the key-role that the immune system cells play in mediating BE/AE, one of the main objectives of SBRT-PATHY approach is to spare these cells from radiation in the PIM-region. Table 3 shows the immunogenic effects of radiation.

Several reports showed that immune system activation with dense lymphocyte infiltration in irradiated tumor sites was associated with a favorable clinical outcome and improved survival [64,67,68,69,70,71,72,73]. However, such evidence was not correlated with improved AE-induction. The reason might be found in the way how conventional radiotherapy has been delivered to the tumor. Assuming that the immune system modulation following tumor irradiation takes place at tumor surface, within the junction where tumor cells meet surrounding antigen-presenting cells and infiltrative lymphocytes, it might be hypothesized that conventional radiotherapy approach that delivers high-dose radiation to this region is probably not an adequate strategy at least for induction of the immune-mediated non-targeted radiation effects. By delivering a full radiation dose to the clinical target volume (CTV) and planning target volume (PTV), which both include significant amount of peritumoral healthy tissue, the loco-regional immune cells will be damaged and killed. While the CTV and PTV are targeted by the mean of conventional whole tumor irradiation and PIM will receive radiation dose same as tumor, irradiation by mean of SBRT-PATHY will maximally spare regions corresponding to PIM. Histologic analysis of patients treated with SBRT-PATHY showed tumor response in terms of necrotic areas ranging from partial (with 25% of vital tumor tissue) to complete response (no vital tumor tissue) at the level of partially irradiated bulky tumors and unirradiated regional tumor sites. Furthermore infiltration of lymphocytes was present in border area with normal tissue, or in vital tumor tissue. Majority of these lymphocytes were T-lymphocytes, with more or less pronounced predomination of CD8+ cytotoxic T-lymphocytes, especially in areas where better response was observed. Shift in the direction of higher numbers of CD8+ lymphocytes was also observed when comparing small pre-treatment biopsies with resection material after therapy. FoxP3 staining and CD20 did not show consistency. Although a small number, thorough analysis of four cases indicate a possible antitumor-directed, radiation-induced activation of the immune system locally and at distance. Furthermore, AIF was highly expressed not only in the partially irradiated primary tumor (SCC), but also in the non-irradiated AC and metastatic lymph nodes in mediastinum. This points to an induction of apoptosis at all sites (irradiated and non-irradiated). Contrary to our and others belief that radiologically confirmed AEs are induced by the direct (cytotoxic) activation of the immune system, our findings are the first to demonstrate an absence of lymphocyte infiltration at abscopal tumor sites, where AIF was highly upregulated. This might be interpreted as radiation-induced activation of an alternative apoptosis pathway through cytochrome C, since AIF is a mitochondrial protein related to the cytochrome C apoptotic pathway [74].

In addition to direct attack by immune cells, cell death-regulating signaling molecules could contribute to tumor cell death at abscopal sites. AIF was detectable by both immunohistochemistry (see above) and PCR and, interestingly, by the latter technique showed even higher expression at non-irradiated abscopal sites than at the irradiated sites. Interferon gamma, IL-6, TNFα, and TRAIL are all secreted signaling molecules that have prominent roles in the regulation of cell death in various cell types, including cancer cells in addition to their immune regulatory functions [75,76,77,78,79]. These signaling molecules have been implicated in bystander and abscopal effects of radiotherapy in preclinical models [80,81]. Their presence in clinical material from abscopal sites as found here could indicate that they may play a part in mediating the systemic anti-tumor response modulated by SBRT-PATHY at PIM and hypoxic segment of partially irradiated bulky cancers.

## 7. Conclusions

In conclusion, adding an immune component in terms of BE/AE to radiation in tumor cell killing may improve the radiotherapy therapeutic ratio in patients that were expected to do poorly otherwise. It seems that the sparing of loco-regional immune cells at the time of an effective tumor-antigen release following the high-dose radiation of massive tumors is capable of inducing immunomodulatory effects of SBRT-PATHY, which may explain the clinical outcomes supported by immunohistochemistry- and gene expression analysis. The future of the radiation-induced immune-mediated non-targeted effects seems to be promising and beyond the conventional treatment approaches, requiring an optimization of radiotherapy in order to increase its immunogenicity. However, further controlled trials and mechanistic investigations in model systems will be required to rigorously dissect and finally confirm the potential role of “soluble abscopal signals” released by tumor microenvironment and tumor itself following partial irradiation. Several studies on SBRT-PATHY approach are ongoing in order to address the mechanisms behind the radiation-hypoxia-induced BE/AE, advantages that carbon-ions might add to this approach in terms of their inverse dose-depth profile, high-LET (linear energy transfer) and RBE (relative biologic effectiveness), as well as the role of PATHY-timing in generating BE/AE (42, 44).

## 8. Patents

Tubin Slavisa M.D. reported an international patent application PCT/EP2019/052164 published as WO 2019/162050. The authors reported no other conflicts of interest.

## Figures and Tables

**Figure 1 cancers-13-00050-f001:**
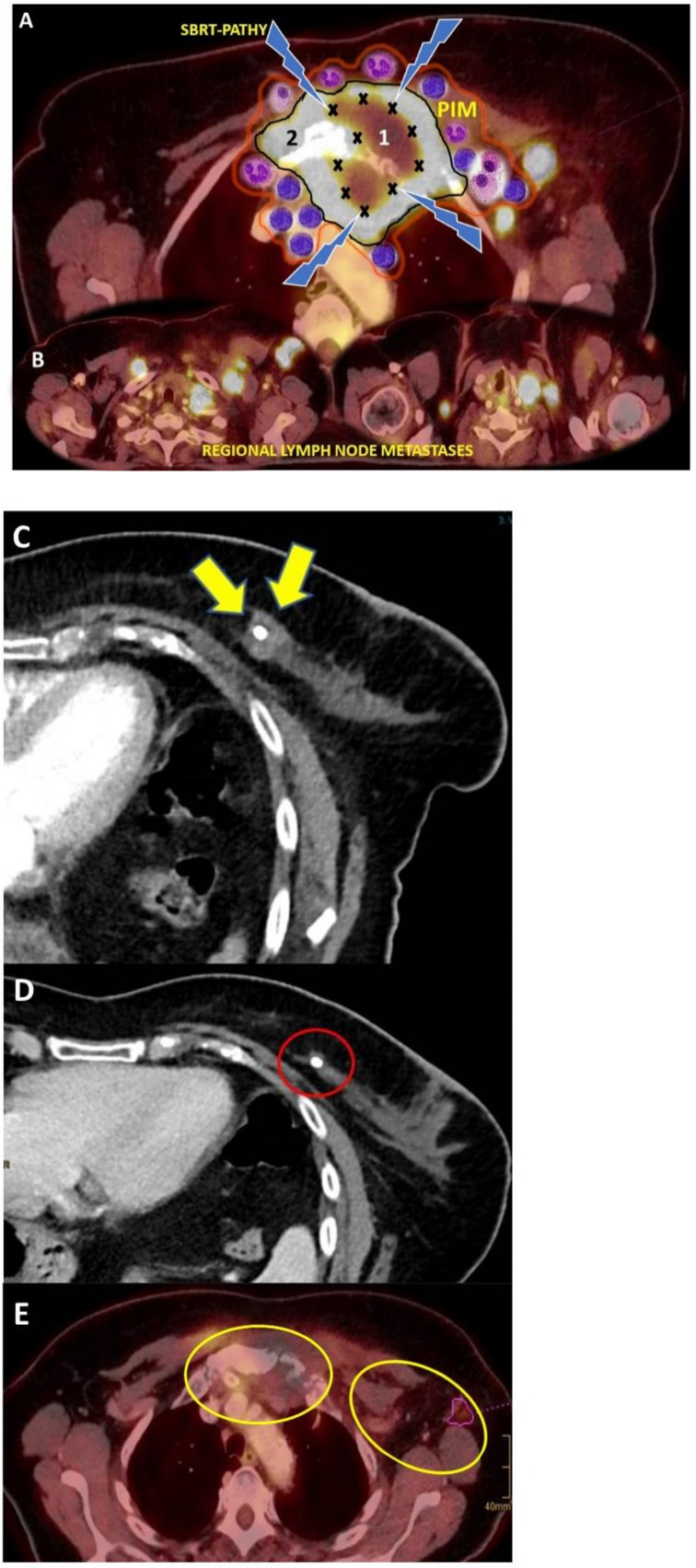
Principle and outcomes of SBRT-PATHY: (**A**) Planning FDG-PET-contrast-enhanced CT shows a large bone metastasis of primary breast cancer infiltrating the nearby soft tissues. Additionally, this 64-years old patient had multiple regional, bilateral lymph node metastases (**B**). After having experienced the disease progression following chemotherapy, this patient was enrolled in the phase I prospective study and submitted to SBRT-PATHY 10 Gy × 3 prescribed to the 70%-isodose line, delivered to the hypovascularized and hypometabolic tumor segment (marked with “X”)-junctional zone between the central necrotic (1) and peripheral hypervascularized and hypermetabolic tumor segment (2) as shown in A. PIM (Peritumoral Immune Microenvironment-red contour) surrounding the tumor surface (black contour) has been maximally spared from radiation in order to preserve its functionality and potential role in mediating the bystander and abscopal effects (**A**). The control CT 2 months after SBRT-PATHY showed complete response of untreated primary breast cancer due to the abscopal effect. (**C**) primary breast cancer (indicated by the yellow arrows and marked by the surgical clip) before SBRT-PATHY; (**D**) complete disappearance of untreated primary breast cancer (indicated by the red ring) after SBRT-PATHY delivered to the bulky bone metastasis. (**E**) PET-CT 5 months following SBRT-PATHY showed complete response of partially irradiated bulky bone metastasis due to the bystander effect, but also of unirradiated regional lymph node metastases because of the abscopal effect (indicated by the yellow contours, in respect to A,B).

**Figure 2 cancers-13-00050-f002:**
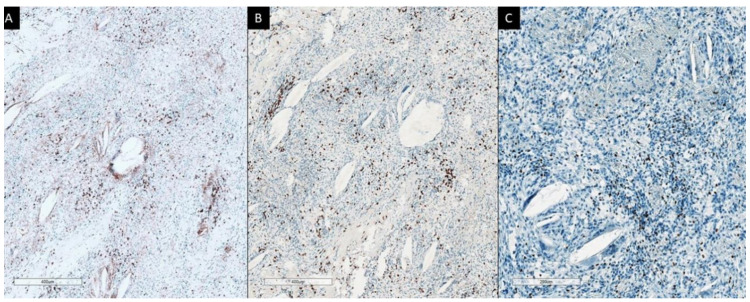
Immunohistochemistry: Histologic presentation of CD4+ (**A**) and CD8+ T-lymphocytes (**B**) in the area adjacent to necrosis of the patient with lung cancer treated with SBRT-PATHY with complete histological response, where slight predominance of later is obvious. FoxP3 positive cells and small cluster are also present (**C**).

**Figure 3 cancers-13-00050-f003:**
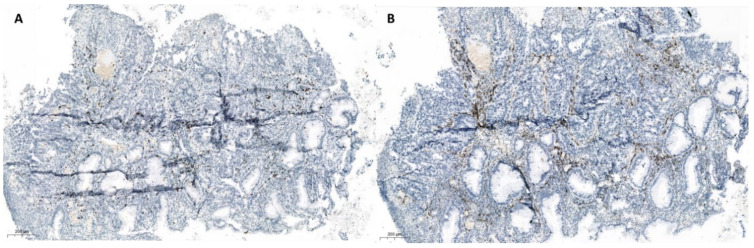
Histologic presentation of a small biopsy of a colon carcinoma with similar amount of CD8 (**A**) and CD4 (**B**) positive lymphocytes.

**Figure 4 cancers-13-00050-f004:**
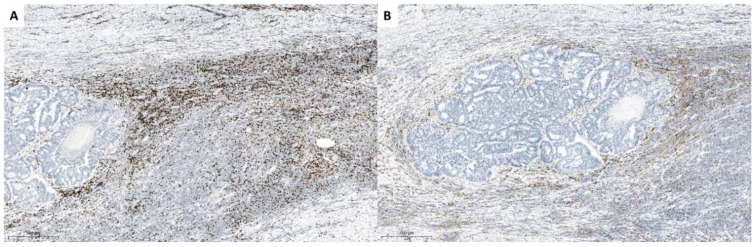
Resection specimen of the same patient as in Figure 3 of a colon carcinoma with predominance of CD8 (**A**) over CD4 (**B**) positive lymphocytes.

**Figure 5 cancers-13-00050-f005:**
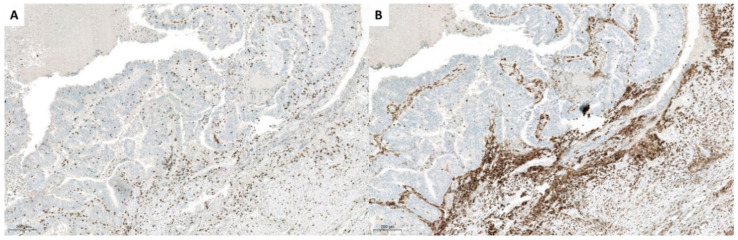
Histologic presentation of a resection specimen of the second patient with colon carcinoma where there are less CD8 (**A**) compared to CD4 (**B**) positive lymphocytes.

**Figure 6 cancers-13-00050-f006:**
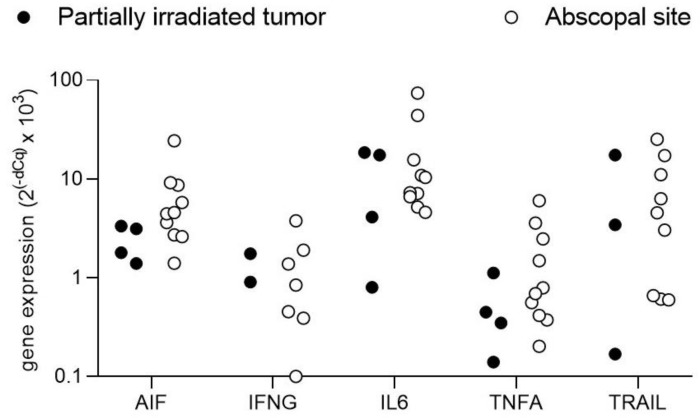
Gene expression analysis: Relative gene expression levels of apoptosis-inducing factor (AIF), interferon gamma (IFNG), interleukin 6 (IL-6), tumor necrosis factor alpha (TNFA) and TNF-related apoptosis-inducing ligand (TRAIL) in partially irradiated tumors (black circles) and abscopal sites (non-irradiated tumors and lymph nodes, empty circles) of four patients treated with SBRT-PATHY. Each circle represents the expression level of the indicated genes analyzed in duplicates in one tissue sample and normalized to the expression of the housekeeping gene GAPDH.

**Table 1 cancers-13-00050-t001:** Treatment characteristics of the selected studies.

Authors (Year of Publication) [Ref.]	Tubin et al. (2017) [21]	Tubin et al. (2019) [37]	Massaccesi * et al. (2019) [38]	Tubin et al. (2019) [39]	Tubin ** et al. (2020) [40]	Tubin et al. (2019) [41]	Tubin *** et al. (2020) [42]
Type of study	Retrospective	Retrospective phase II	Retrospective case series (re-irradiation)	Retrospective	Retrospective	Prospective	Prospective phase I
Number of patient sunderwent SBRT-PATHY	7	20	8	23	3	8	20
Median follow up (months)	6 (2–9)	13 (4–27)	7 (1–15)	9.4 (4–20)	5.3 (3–7)	11.8 (4–22)	9 (4–12)
Local control (bystander effect)	100%	95%	83%	96%	67%	75%	73%
Abscopal response	28.6%	45%	Not evaluable	52%	Not evaluable	50%	47%
Symptom relief	100%	80%	100%	96%	67%	88%	82%
Treated symptoms	Dyspnea, pain.	Dyspnea, pain, cough, hemoptysis.	Pain, bleeding	Dyspnea, pain, cough.	Pain, Dysphagia.	Dyspnea, pain, cough.	Dyspnea, pain, cough, haemoptysis, edema-extremities, dysphonia.
Toxicity	none	Fatigue G1 (15%)	none	none	none	none	Fatigue G1 (20%)
Hematological toxicity/leucopenia	none	none	none	none	none	none	none
Median total dose/dose-fraction (Gy)	10/10	10–30/10	10/10	10–30/10	36/12	30/10	30/10

* abstract (presented at ASTRO annual meeting 2019). ** unpublished data (ongoing phase I study on the use of carbon-ions for –PATHY approach). *** unpublished data (ongoing phase I proof of principle trial).

**Table 2 cancers-13-00050-t002:** Immunohistochemistry and gene-expression findings following SBRT-PATHY.

		Gene-Expression Findings			
Tumor Site	Immunohistochemistry Findings	TNF	IL6	TRAIL	IFNG
Patient 1. SCC lung (partially irradiated, bystander site)	80% necrosis, 20% viable tumor cells, dense aggregation of lymphocytes in PIM, focal accumulation of CD20+ B-lymphocytes, dense infiltration of CD3+ T-lymphocytes (CD8+), high number of CD14+ myeloid-derived suppressor cells, small number of CD15+ myeloid-derived suppressor cells, CD56+NK cells were not detectable; AIF was highly expressed.	+	+	+	+
AC lung (unirradiated, abscopal site)	80% necrosis, 20% viable tumor cells, no aggregation of lymphocytes, CD20+ B-lymphocytes absent, CD3+ T-lymphocytes present (predominantly CD4+), low number of CD14+ myeloid-derived suppressor cells, small number of CD15+ myeloid-derived suppressor cells, CD56+NK cells were not detectable; AIF was highly expressed.	+	+	+	−
Metastatic lymph nodes hilus/mediastinum (unirradiated, abscopal site)	100% necrosis, no viable tumor cells, present some CD20+ B-lymphocytes, prevalent infiltration of CD3+ T-lymphocytes (CD8+), high number of CD14+ myeloid-derived suppressor cells, small number of CD15+ myeloid-derived suppressor cells, CD56+NK cells were not detectable; AIF was highly expressed.	+	+	+	+
Patient 2. AC lung (partially irradiated, bystander site)	100% necrosis, no viable tumor cells, dense lymphocytic infiltrates in PIM-region, foamy macrophages and multinucleated giant cells, all lymphocytes were CD3+ (CD8+ T-lymphocytes), CD4:CD8 ratio 1:2, higher number of FoxP3 positive cells, CD20+ and PD-L1 were negative; AIF was highly expressed.	+	+	+	−
AC lung (prior SBRT-PATHY)	Majority of lymphocytes were CD3+ with equal distribution of CD4+ and CD8+ cells, FoxP3 also stained isolated cells, CD20+ and PD-L1 negative.	NA			
Metastatic lymph nodes hilus/mediastinum (unirradiated, abscopal site)	100% necrosis, no viable tumor cells; AIF was highly expressed.	+	+	+	−
Patient 3. AC rectum (partially irradiated, bystander site)	75% necrosis, 25% viable tumor cells, lymphocytic reaction with focal fibrosis, CD3+ T lymphocytes formed the majority of cells, predominance of CD8+ (CD4:CD8 ratio 1:2), CD20+ cells within vital tumor tissue, FoxP3 was negative, PD-L1 negative, very strong cytoplasmatic expression of CD14+ in PIM-region; AIF was highly expressed.	+	+	+	+
AC rectum (prior SBRT-PATHY)	More CD4+ T-lymphocytes than CD8+ T-lymphocytes, and few cells were also CD20+ and FoxP3+. PD-L1 negative.	NA			
Metastatic lymph nodes pelvis/mesorectum (unirradiated, abscopal site)	100% necrosis, no viable tumor cells. AIF was highly expressed.	+	+	+	+
AC caecum (unirradiated, abscopal site)	AIF was highly expressed.	+	+	+	+
Patient 4. AC rectum (partially irradiated, bystander site)	75% necrosis, 25% viable tumor cells, lymphocytic reaction with focal fibrosis, CD3+ T lymphocytes formed the majority of cells, more or less the same number of CD4+ and CD8+ T-lymphocytes (CD4:CD8 ratio 1:1), FoxP3 was negative, PD-L1 negative. Very strong cytoplasmatic expression of CD14+ in PIM-region; AIF was highly expressed.	+	+	−	−
AC rectum (prior SBRT-PATHY)	Similar number of CD4+ and CD8+ cells and few cells were also CD20+ and FoxP3+. PD-L1 negative. Very strong cytoplasmatic expression of CD14.	NA			
Metastatic lymph nodes pelvis/mesorectum (unirradiated, abscopal site)	100% necrosis, no viable tumor cells. AIF was highly expressed.	+	+	+	+

SCC-squamous cell carcinoma, AC-adenocarcinoma, PIM-peritumoral immune microenvironment, NA-not applicable, AIF-apoptosis-inducing factor, IFNG-interferon gamma, IL6-interleukine 6, TNF-tumor necrosis factor, TRAIL-TNF-related apoptosis-inducing ligand.

**Table 3 cancers-13-00050-t003:** Immunogenic effects of radiotherapy [60,61,62,63,64,65,66].

Immunostimulatory Effects	
Calreticulin translocation to the surface of tumor cells (“eat me” signal) *	Increased tumor cells phagocytosis Promotes pro-inflammatory cytokines release from APCs
Release of HMGB1 protein (“danger signal”) *	DC migration and maturation (increase in efficiency of antigen processing and presentation) Release of pro-inflammatory cytokines and chemokines from APCs
Release of ATP *	Release of pro-inflammatory cytokines from APCs (priming of IFN-γ-producing cytotoxic CD8+ T cells)
HSP increase (membrane-bound expression and extracellular release) *	Stimulate innate and adaptive immune responses
Decrease of CD47 surface expression (“do not-eat-me” signal)	Increase tumor cells phagocytosis
Accumulation of cytosolic DNA in irradiated tumor cells *	Activation of the cGAS/STING pathway and production of type I IFNs and other pro-inflammatory cytokines (APCs maturation, cross-presentation and T cell recruitment)
Smac release from mitochondria	Increase tumor cells sensitivity to granzyme-induced apoptosis
Generation of novel peptides and increase of the pool of intracellular peptides presented	Increase the anti-tumor immune response
Increased MHC-I expression (critical for antigen recognition by CD8+ TCRs)	Enhance recognition and killing of cancer cells by cytotoxic T cells
Increase of NKG2D ligands, co-stimulatory molecules (e.g., CD80) and adhesion molecules (e.g., ICAM-1, E-selectin) on tumor cells	Enhance recognition and killing of cancer cells by cytotoxic lymphocytes
Upregulation of “death receptors” (e.g., FAS/CD95)	Enhance recognition and killing of cancer cells by cytotoxic lymphocytes
Release of chemokines (e.g., CXCL9, CXCL10, CXCL16,), increase of adhesion molecules on the vascular endothelium (e.g., VCAM-1), normalization of the tumor vasculature	Facilitate the recruitment of effector T-cells to the tumor site
**Immunosuppressive Effects**	
Upregulation of PDL-1 on cancer cells	Inhibit CTL-mediated tumor killing
Accumulation of regulatory T cells (related to intrinsic higher radio-resistance and increase of immunosuppressive mediators and cytokines induced by radiation)	Immunosuppression
Accumulation of immunosuppressive myeloid cells (N2 neutrophils, M2 macrophages, MDSCs) secondary to the increase of CSF-1, SDF-1, CCL2 induced by radiation	Immunosuppression
Induction of TGF-beta secretion	Multiple immunosuppressive effects
Upregulation of the transcription of HIF-1α	Multiple immunosuppressive effects
Upregulation of adenosine	Multiple immunosuppressive effects
Killing of tumor-infiltrating immune cells (e.g., lymphocytes, APCs)	Immunosuppression

APCs; antigen-presenting cells; ATP, adenosine triphosphate; CSF-1, colony-stimulating factor 1; DC, Dendritic cells; cGAS, GMP-AMP synthase; HIF, Hypoxia-Inducible Factor; HSP, Heat-Shock Proteins, HMGB1, High Mobility Group Box 1; IL, Interleukin; IFN, Interferon; MDSC, myeloid-derived suppressor cell; NKG2D, natural killer group 2D; PDL-1, programmed death ligand-1; SDF-1, stromal cell derived factor-1; STING, STimulator of INterferon Genes; TCR, T-cell receptor.* cellular phenomena related to the “immunogenic cell death” of the tumor cell.

## Data Availability

Data sharing is not applicable to this review article.

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
