# Peer review of "Shifting the Immune-Suppressive to Predominant Immune-Stimulatory Radiation Effects by SBRT-PArtial Tumor Irradiation Targeting HYpoxic Segment (SBRT-PATHY)"

_cancers, 2020, doi:10.3390/cancers13010050_

Round 1

Reviewer 1 Report

Thank you so much for your response. I understand that you already have published several articles with the clinical data of SBRT-PATHY. However, most of them are retrospective or non-randomized prospective study with small number of patients. In that mean, I think the evidence of clinical outcome of SBRT-PATHY is still preliminary.
I agree that this newer technique is a great promise clinically. Therefore, at this stage, I think that it is better to state that clinical evidence is still limited in the section of "3. Clinical outcomes".

Author Response

Dear Reviewer 1,

Thank you. We appreciate all your comments and suggestions made in order to improve the quality of our paper. That saying, we have added, like you proposed, to the section "3. Clinical outcomes" the following phrase that addresses your suggestion: "Clinical evidence on the use of this novel approach is still limited to mainly retrospective or non-randomized prospective study data, with small number of patients."(lines 224-225).

Reviewer 2 Report

Without to be convincing, authors replied to all comments

Author Response

Dear Reviewer 2.,

I would like to thank you for all your suggestions and comments made in order to improve the quality of our paper.

This manuscript is a resubmission of an earlier submission. The following is a list of the peer review reports and author responses from that submission.

Round 1

Reviewer 1 Report

The authors described this article as a review about abscopal and bystander effects using a technic of partial irradiation of the tumor. The project was relevant but failed by the presentation of the manuscript.

1/ authors reported their own translational results which were not completely published and without figures and specific results allowing to criticize of the material and methods and consequently the reported results.

2/ furthermore, they use this manuscript to introduce their results of their very disputable clinical "trial". As they explained in the text, patient gave their agreement to receive partial irradiation but without ethic approval of the protocol. This method is not acceptable. Moreover, the irradiation technic as presented is highly disputable and results cannot lead to the conclusions of the authors.

3/ discussion is not acceptable in the form and some paragraphs "This explanation is analogous to the menstrual cycle..." are completely out of the focus of manuscript goal.

If authors hope to publish this article the must stay in the subject, with scientific, ethical publications as reference. The analysis of the data must be focused on the cancer fields and should avoid pseudo-philosophical or pseudo-scientific analogies.

Reviewer 2 Report

This review showed preclinical findings suggesting that the high-dose irradiation of hypoxic tumor cells generates stronger bystander effect (BE) and abscopal effect (AE) and preliminary some clinical evidences. And also, the authors discusses the current evidence on the safety and efficacy of SBRT-PATHY in the treatment of unresectable hypoxic bulky tumors and its bystander and abscopal immunomodulatory potential.

This review is very interesting in consideration of future possibility of new approach combining immuno-modulating agents and high-precision radiotherapy. I think that SBRT-PATHY is an unique method of them. However, the evidence of clinical outcomes seems to be too preliminary to discuss the efficacy of SBRT-PATHY. I don't think there's enough evidence yet to mention its clinical usefulness or validity.